# LARGE LANGUAGE MODELS CANNOT SELF-CORRECT REASONING YET

**Jie Huang**[1,2][*] **Xinyun Chen**[1][*] **Swaroop Mishra**[1] **Huaixiu Steven Zheng**[1] **Adams Wei Yu**[1]
**Xinying Song**[1] **Denny Zhou**[1]

[1]Google DeepMind   [2]University of Illinois at Urbana-Champaign

jeffhj@illinois.edu, {xinyunchen, dennyzhou}@google.com

## ABSTRACT

Large Language Models (LLMs) have emerged as a groundbreaking technology with their unparalleled text generation capabilities across various applications. Nevertheless, concerns persist regarding the accuracy and appropriateness of their generated content. A contemporary methodology, *self-correction*, has been proposed as a remedy to these issues. Building upon this premise, this paper critically examines the role and efficacy of self-correction within LLMs, shedding light on its true potential and limitations. Central to our investigation is the notion of *intrinsic self-correction*, whereby an LLM attempts to correct its initial responses based solely on its inherent capabilities, without the crutch of external feedback. In the context of reasoning, our research indicates that LLMs struggle to self-correct their responses without external feedback, and at times, their performance even degrades after self-correction. Drawing from these insights, we offer suggestions for future research and practical applications in this field.

## 1 INTRODUCTION

The rapid advancements in the domain of artificial intelligence have ushered in the era of Large Language Models (LLMs). These models, characterized by their expansive parameter counts and unparalleled capabilities in text generation, have showcased promising results across a multitude of applications (Chowdhery et al., 2023; Anil et al., 2023; OpenAI, 2023, *inter alia*). However, concerns about their accuracy, reasoning capabilities, and the safety of their generated content have drawn significant attention from the community (Bang et al., 2023; Alkaissi & McFarlane, 2023; Zheng et al., 2023; Shi et al., 2023; Carlini et al., 2021; Huang et al., 2022; Shao et al., 2023; Li et al., 2023; Wei et al., 2023; Zhou et al., 2023b; Zou et al., 2023, *inter alia*).

Amidst this backdrop, the concept of "self-correction" has emerged as a promising solution, where LLMs refine their responses based on feedback to their previous outputs (Madaan et al., 2023; Welleck et al., 2023; Shinn et al., 2023; Kim et al., 2023; Bai et al., 2022; Ganguli et al., 2023; Gao et al., 2023; Paul et al., 2023; Chen et al., 2023b; Pan et al., 2023, *inter alia*). However, the underlying mechanics and efficacy of self-correction in LLMs remain underexplored. A fundamental question arises: If an LLM possesses the ability to self-correct, why doesn't it simply offer the correct answer in its initial attempt? This paper delves deeply into this paradox, critically examining the self-correction capabilities of LLMs, with a particular emphasis on reasoning (Wei et al., 2022; Zhou et al., 2023b; Huang & Chang, 2023).

To study this, we first define the concept of *intrinsic self-correction*, a scenario wherein the model endeavors to rectify its initial responses based solely on its inherent capabilities, without the crutch of external feedback. Such a setting is crucial because high-quality external feedback is often unavailable in many real-world applications. Moreover, it is vital to understand the intrinsic capabilities of LLMs. Contrary to the optimism surrounding self-correction (Madaan et al., 2023; Kim et al., 2023; Shinn et al., 2023; Pan et al., 2023, *inter alia*), our findings indicate that LLMs struggle to self-correct their reasoning in this setting. In most instances, the performance after self-correction

---

[*]Equal contribution.

even deteriorates. This observation is in contrast to prior research such as Kim et al. (2023); Shinn et al. (2023). Upon closer examination, we observe that the improvements in these studies result from using oracle labels to guide the self-correction process, and the improvements vanish when oracle labels are not available.

Besides the reliance on oracle labels, we also identify other issues in the literature regarding measuring the improvement achieved by self-correction. First, we note that self-correction, by design, utilizes multiple LLM responses, thus making it crucial to compare it to baselines with equivalent inference costs. From this perspective, we investigate multi-agent debate (Du et al., 2023; Liang et al., 2023) as a means to improve reasoning, where multiple LLM instances (can be multiple copies of the same LLM) critique each other's responses. However, our results reveal that its efficacy is no better than self-consistency (Wang et al., 2022) when considering an equivalent number of responses, highlighting the limitations of such an approach.

Another important consideration for self-correction involves prompt design. Specifically, each self-correction process involves designing prompts for both the initial response generation and the self-correction steps. Our evaluation reveals that the self-correction improvement claimed by some existing work stems from the sub-optimal prompt for generating initial responses, where self-correction corrects these responses with more informative instructions about the initial task in the feedback prompt. In such cases, simply integrating the feedback into the initial instruction can yield better results, and self-correction again decreases performance.

In light of our findings, we provide insights into the nuances of LLMs' self-correction capabilities and initiate discussions to encourage future research focused on exploring methods that can genuinely correct reasoning.

## 2 BACKGROUND AND RELATED WORK

With the LLM evolution, the notion of self-correction gained prominence. The discourse on self-correction pivots around whether these advanced models can recognize the correctness of their outputs and provide refined answers (Bai et al., 2022; Madaan et al., 2023; Welleck et al., 2023, *inter alia*). For example, in the context of mathematical reasoning, an LLM might initially solve a complex problem but make an error in one of the calculation steps. In an ideal self-correction scenario, the model is expected to recognize the potential mistake, revisit the problem, correct the error, and consequently produce a more accurate solution.

Yet, the definition of "self-correction" varies across the literature, leading to ambiguity. A pivotal distinction lies in the source of feedback (Pan et al., 2023), which can purely come from the LLM, or can be drawn from external inputs. Internal feedback relies on the model's inherent knowledge and parameters to reassess its outputs. In contrast, external feedback incorporates inputs from humans, other models (Wang et al., 2023b; Paul et al., 2023, *inter alia*), or external tools and knowledge sources (Gou et al., 2023; Chen et al., 2023b; Olausson et al., 2023; Gao et al., 2023, *inter alia*).

In this work, we focus on examining the self-correction capability of LLMs for reasoning. Reasoning is a fundamental aspect of human cognition, enabling us to understand the world, draw inferences, make decisions, and solve problems. To enhance the reasoning performance of LLMs, Kim et al. (2023); Shinn et al. (2023) use oracle labels about the answer correctness to guide the self-correction process. However, in practice, high-quality external feedback such as answer correctness is often unavailable. For effective self-correction, the ability to judge the correctness of an answer is crucial and should ideally be performed by the LLM itself. Consequently, our focus shifts to self-correction without any external or human feedback. We term this setting **intrinsic self-correction**. For brevity, unless explicitly stated otherwise (e.g., self-correction with oracle feedback), all references to "self-correction" in the remainder of this paper pertain to intrinsic self-correction.

In the following sections, we will evaluate a variety of existing self-correction techniques. We demonstrate that existing techniques actually decrease reasoning performance when oracle labels are not used (Section 3), perform worse than methods without self-correction when utilizing the same number of model responses (Section 4), and lead to less effective outcomes when using informative prompts for generating initial responses (Section 5). We present an overview of issues in the evaluation setups of previous LLM self-correction works in Table 1, with detailed discussions in the corresponding sections.

Table 1: Summary of issues in previous LLM self-correction evaluation.

| Method | Issue |
|---|---|
| RCI (Kim et al., 2023); Reflexion (Shinn et al., 2023) | Use of oracle labels (Section 3) |
| Multi-Agent Debate (Du et al., 2023) | Unfair comparison to self-consistency (Section 4) |
| Self-Refine (Madaan et al., 2023) | Sub-optimal prompt design (Section 5) |

## 3  LLMs Cannot Self-Correct Reasoning Intrinsically

In this section, we evaluate existing self-correction methods and compare their performance with and without oracle labels regarding the answer correctness.

### 3.1  Experimental Setup

**Benchmarks.** We use datasets where existing self-correction methods with oracle labels have demonstrated significant performance improvement, including

- **GSM8K** (Cobbe et al., 2021): GSM8K comprises a test set of 1,319 linguistically diverse grade school math word problems, curated by human problem writers. There is a notable improvement of approximately 7% as evidenced by Kim et al. (2023) after self-correction.
- **CommonSenseQA** (Talmor et al., 2019): This dataset offers a collection of multi-choice questions that test commonsense reasoning. An impressive increase of around 15% is showcased through the self-correction process, as demonstrated by Kim et al. (2023). Following Kojima et al. (2022); Kim et al. (2023), we utilize the dev set for our evaluation, which encompasses 1,221 questions.
- **HotpotQA** (Yang et al., 2018): HotpotQA is an open-domain multi-hop question answering dataset. Shinn et al. (2023) demonstrate significant performance improvement through self-correction. We test models' performance in a closed-book setting and evaluate them using the same set as Shinn et al. (2023). This set contains 100 questions, with exact match serving as the evaluation metric.

**Test Models and Setup.** We first follow Kim et al. (2023); Shinn et al. (2023) to evaluate the performance of self-correction with oracle labels, using GPT-3.5-Turbo (`gpt-3.5-turbo-0613`) and GPT-4 accessed on 2023/08/29. For intrinsic self-correction, to provide a more thorough analysis, we also evaluate GPT-4-Turbo (`gpt-4-1106-preview`) and Llama-2 (`Llama-2-70b-chat`) (Touvron et al., 2023). For GPT-3.5-Turbo, we employ the full evaluation set. For other models, to reduce the cost, we randomly sample 200 questions for each dataset (100 for HotpotQA) for testing. We prompt the models to undergo a maximum of two rounds of self-correction. We use a temperature of 1 for GPT-3.5-Turbo and GPT-4, and a temperature of 0 for GPT-4-Turbo and Llama-2, to provide evaluation across different decoding algorithms.

**Prompts.** Following Kim et al. (2023); Shinn et al. (2023), we apply a three-step prompting strategy for self-correction: 1) prompt the model to perform an initial generation (which also serves as the results for Standard Prompting); 2) prompt the model to review its previous generation and produce feedback; 3) prompt the model to answer the original question again with the feedback.

For our experiments, we mostly adhere to the prompts from the source papers. For GSM8K and CommonSenseQA, we integrate format instructions into the prompts of Kim et al. (2023) to facilitate a more precise automatic evaluation (detailed prompts can be found in Appendix A). For HotpotQA, we use the same prompt as Shinn et al. (2023). We also assess the performance of various self-correction prompts for intrinsic self-correction. For example, we use "*Assume that this answer could be either correct or incorrect. Review the answer carefully and report any serious problems you find.*" as the default feedback prompt for the evaluation on GPT-4-Turbo and Llama-2.

### 3.2  Results

**Self-Correction with Oracle Labels.** Following previous works (Kim et al., 2023; Shinn et al., 2023), we use the correct label to determine when to stop the self-correction loop. This means we

Table 2: Results of GPT-3.5 and GPT-4 on reasoning benchmarks with oracle labels.

|  |  | GSM8K | CommonSenseQA | HotpotQA |
|---|---|---|---|---|
| GPT-3.5 | Standard Prompting | 75.9 | 75.8 | 26.0 |
|  | Self-Correct (Oracle) | 84.3 | 89.7 | 29.0 |
| GPT-4 | Standard Prompting | 95.5 | 82.0 | 49.0 |
|  | Self-Correct (Oracle) | 97.5 | 85.5 | 59.0 |

Table 3: Results of GPT-3.5 and GPT-4 on reasoning benchmarks with intrinsic self-correction.

|  |  | # calls | GSM8K | CommonSenseQA | HotpotQA |
|---|---|---|---|---|---|
| GPT-3.5 | Standard Prompting | 1 | **75.9** | **75.8** | **26.0** |
|  | Self-Correct (round 1) | 3 | 75.1 | 38.1 | 25.0 |
|  | Self-Correct (round 2) | 5 | 74.7 | 41.8 | 25.0 |
| GPT-4 | Standard Prompting | 1 | **95.5** | **82.0** | **49.0** |
|  | Self-Correct (round 1) | 3 | 91.5 | 79.5 | **49.0** |
|  | Self-Correct (round 2) | 5 | 89.0 | 80.0 | 43.0 |

utilize the ground-truth label to verify whether each step's generated answer is correct. If the answer is already correct, no (further) self-correction will be performed. Table 2 summarizes the results of self-correction under this setting, showcasing significant performance improvements, consistent with the findings presented in Kim et al. (2023); Shinn et al. (2023).

However, these results require careful consideration. For reasoning tasks, like solving mathematical problems, the availability of oracle labels seems counter-intuitive. If we are already in possession of the ground truth, there seems to be little reason to deploy LLMs for problem-solving. Therefore, the results can only be regarded as indicative of an oracle's performance.

**Intrinsic Self-Correction.** Per the above discussion, performance improvements achieved using oracle labels do not necessarily reflect true self-correction ability. Therefore, we turn our focus to the results in the *intrinsic self-correction* setting as defined in Section 2. To achieve this, we eliminate the use of labels, requiring LLMs to independently determine when to stop the self-correction process, i.e., whether to retain their previous answers.

Tables 3 and 4 report the accuracies and the number of model calls. We observe that, after self-correction, the accuracies of all models drop across all benchmarks.

To provide a more comprehensive assessment, we also design several different self-correction prompts to determine if there are better prompts that could enhance reasoning performance. Nonetheless, as shown in Tables 5 and 6, without the use of oracle labels, self-correction consistently results in a decrease in performance.

### 3.3 WHY DOES THE PERFORMANCE NOT INCREASE, BUT INSTEAD DECREASE?

**Empirical Analysis.** Figure 1 summarizes the results of changes in answers after two rounds of self-correction, with two examples of GPT-3.5 illustrated in Figure 2. For GSM8K, 74.7% of the time, GPT-3.5 retains its initial answer. Among the remaining instances, the model is more likely to modify a correct answer to an incorrect one than to revise an incorrect answer to a correct one. ***The fundamental issue is that LLMs cannot properly judge the correctness of their reasoning.*** For CommonSenseQA, there is a higher chance that GPT-3.5 alters its answer. The primary reason for this is that false answer options in CommonSenseQA often appear somewhat relevant to the question, and using the self-correction prompt might bias the model to choose another option, leading to a high "correct $\Rightarrow$ incorrect" ratio. Similarly, Llama-2 also frequently converts a correct answer into an incorrect one. Compared to GPT-3.5 and Llama-2, both GPT-4 and GPT-4-Turbo are more likely to retain their initial answers. This may be because GPT-4 and GPT-4-Turbo have higher confidence

Table 4: Results of GPT-4-Turbo and Llama-2 with intrinsic self-correction.

|  |  | # calls | GSM8K | CommonSenseQA |
|---|---|---|---|---|
| GPT-4-Turbo | Standard Prompting | 1 | **91.5** | **84.0** |
|  | Self-Correct (round 1) | 3 | 88.0 | 81.5 |
|  | Self-Correct (round 2) | 5 | 90.0 | 83.0 |
| Llama-2 | Standard Prompting | 1 | **62.0** | **64.0** |
|  | Self-Correct (round 1) | 3 | 43.5 | 37.5 |
|  | Self-Correct (round 2) | 5 | 36.5 | 36.5 |

Table 5: Results of GPT-4-Turbo with different feedback prompts.

|  | # calls | GSM8K | CommonSenseQA |
|---|---|---|---|
| Standard Prompting | 1 | 91.5 | 84.0 |

*Feedback Prompt*: Assume that this answer could be either correct or incorrect. Review the answer carefully and report any serious problems you find.

|  | # calls | GSM8K | CommonSenseQA |
|---|---|---|---|
| Self-Correct (round 1) | 3 | 88.0 | 81.5 |
| Self-Correct (round 2) | 5 | 90.0 | 83.0 |

*Feedback Prompt*: Review your previous answer and determine whether it's correct. If wrong, find the problems with your answer.

|  | # calls | GSM8K | CommonSenseQA |
|---|---|---|---|
| Self-Correct (round 1) | 3 | 90.0 | 74.5 |
| Self-Correct (round 2) | 5 | 90.0 | 81.0 |

*Feedback Prompt*: Verify whether your answer is correct, and provide an explanation.

|  | # calls | GSM8K | CommonSenseQA |
|---|---|---|---|
| Self-Correct (round 1) | 3 | 91.0 | 81.5 |
| Self-Correct (round 2) | 5 | 91.0 | 83.5 |

Table 6: Results of Llama-2 with different feedback prompts.

|  | # calls | GSM8K | CommonSenseQA |
|---|---|---|---|
| Standard Prompting | 1 | 62.0 | 64.0 |

*Feedback Prompt*: Assume that this answer could be either correct or incorrect. Review the answer carefully and report any serious problems you find.

|  | # calls | GSM8K | CommonSenseQA |
|---|---|---|---|
| Self-Correct (round 1) | 3 | 43.5 | 37.5 |
| Self-Correct (round 2) | 5 | 36.5 | 36.5 |

*Feedback Prompt*: Review your previous answer and determine whether it's correct. If wrong, find the problems with your answer.

|  | # calls | GSM8K | CommonSenseQA |
|---|---|---|---|
| Self-Correct (round 1) | 3 | 46.5 | 26.0 |
| Self-Correct (round 2) | 5 | 30.5 | 37.0 |

*Feedback Prompt*: Verify whether your answer is correct, and provide an explanation.

|  | # calls | GSM8K | CommonSenseQA |
|---|---|---|---|
| Self-Correct (round 1) | 3 | 58.0 | 24.0 |
| Self-Correct (round 2) | 5 | 41.5 | 43.0 |

in their initial answers, or because they are more robust and thus less prone to being biased by the self-correction prompt.[1]

---

[1] We omit the analysis on HotpotQA because the sample size used in the source paper is quite small, which may not produce meaningful statistics.

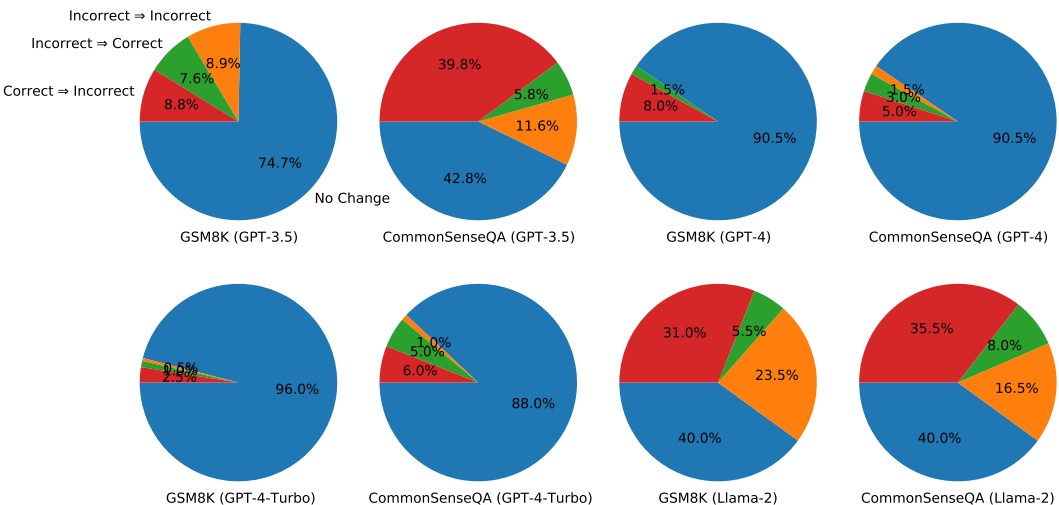

Figure 1: Analysis of the changes in answers after two rounds of self-correction. *No Change*: The answer remains unchanged; *Correct ⇒ Incorrect*: A correct answer is changed to an incorrect one; *Incorrect ⇒ Correct*: An incorrect answer is revised to a correct one; *Incorrect ⇒ Incorrect*: An incorrect answer is altered but remains incorrect.

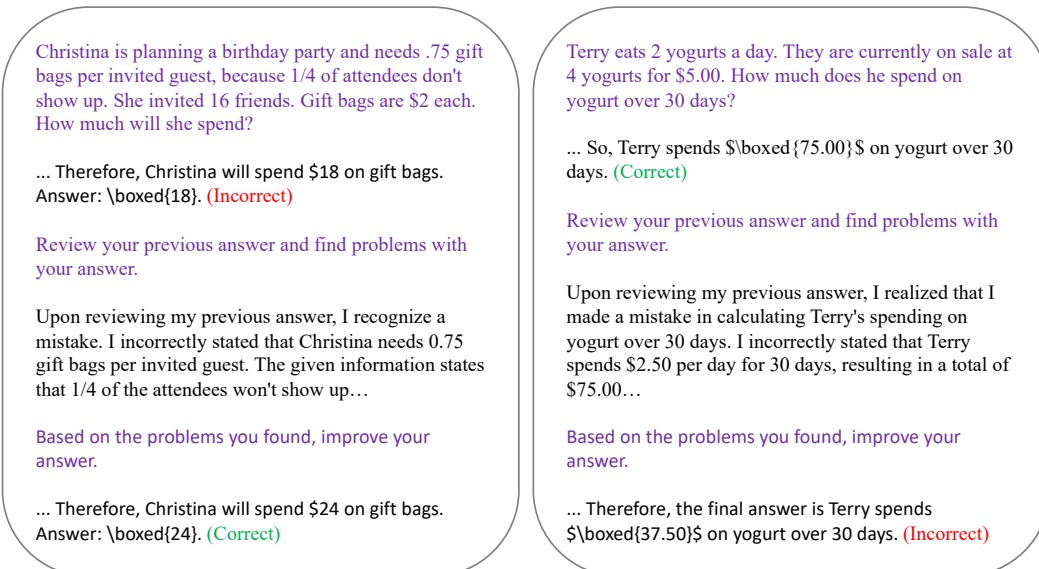

Figure 2: Examples on GSM8K with GPT-3.5. *Left*: successful self-correction; *Right*: failed self-correction. Full prompts and responses can be viewed in Figures 3 and 4 of Appendix A.

Let's take another look at the results presented in Table 2. These results use ground-truth labels to prevent the model from altering a correct answer to an incorrect one. However, ***determining how to prevent such mischanges is, in fact, the key to ensuring the success of self-correction.***

**Intuitive Explanation.** If the model is well-aligned and paired with a thoughtfully designed initial prompt, the initial response should already be optimal relative to the prompt and the specific decoding algorithm. Introducing feedback can be viewed as adding an additional prompt, potentially skewing the model towards generating a response that is tailored to this combined input. In an intrinsic self-correction setting, on the reasoning tasks, this supplementary prompt may not offer any extra advantage for answering the question. In fact, it might even bias the model away from producing an optimal response to the initial prompt, resulting in a performance drop.

Table 7: Results of multi-agent debate and self-consistency.

|  | # responses | GSM8K |
|---|---|---|
| Standard Prompting | 1 | 76.7 |
| Self-Consistency | 3 | 82.5 |
| Multi-Agent Debate (round 1) | 6 | 83.2 |
| Self-Consistency | 6 | 85.3 |
| Multi-Agent Debate (round 2) | 9 | 83.0 |
| Self-Consistency | 9 | **88.2** |

Table 8: Results of Constrained Generation.

|  | # calls | CommonGen-Hard |
|---|---|---|
| Standard Prompting* | 1 | 44.0* |
| Self-Correct* | 7 | 67.0* |
| Standard Prompting* | 1 | 53.0 |
| Self-Correct* | 7 | 61.1 |
| Standard Prompting (ours) | 1 | **81.8** |
| Self-Correct* | 7 | 75.1 |

\* Prompts and results from Madaan et al. (2023).

## 4 MULTI-AGENT DEBATE DOES NOT OUTPERFORM SELF-CONSISTENCY

Another potential approach for LLMs to self-correct their reasoning involves allowing the models to critique and debate through multiple model calls (Du et al., 2023; Liang et al., 2023; Chen et al., 2023a). Du et al. (2023) implement a multi-agent debate method by leveraging multiple instances of a single ChatGPT model and demonstrate significant improvements on reasoning tasks. We adopt their method to test performance on GSM8K. For an unbiased implementation, we use the exact same prompt as Du et al. (2023) and replicate their experiment with the `gpt-3.5-turbo-0301` model, incorporating 3 agents and 2 rounds of debate. The only distinction is that, to reduce result variance, we test on the complete test set of GSM8K, compared to their usage of 100 examples. For reference, we also report the results of self-consistency (Wang et al., 2022), which prompts models to generate multiple responses and performs majority voting to select the final answer.

Table 7 presents the results. The results indicate that both multi-agent debate and self-consistency achieve significant improvements over standard prompting. However, when comparing multi-agent debate to self-consistency, we observe that the performance of multi-agent is only slightly better than that of self-consistency with the same number of agents (3 responses, the baseline also compared in Du et al. (2023)). Furthermore, for self-consistency with an equivalent number of responses, multi-agent debate significantly underperforms simple self-consistency using majority voting.

In fact, rather than labeling the multi-agent debate as a form of "debate" or "critique", it is more appropriate to perceive it as a means to achieve "consistency" across multiple model generations. Fundamentally, its concept mirrors that of self-consistency; the distinction lies in the voting mechanism, whether voting is model-driven or purely based on counts. The observed improvement is evidently not attributed to "self-correction", but rather to "self-consistency". If we aim to argue that LLMs can self-correct reasoning through multi-agent debate, it is preferable to exclude the effects of selection among multiple generations.

## 5 PROMPT DESIGN ISSUES IN SELF-CORRECTION EVALUATION

In Section 3, we observe that although self-correction decreases reasoning performance with all types of feedback prompts we have evaluated, performance varies with different feedback prompts. In this section, we further emphasize the importance of proper prompt design in generating initial LLM responses to fairly measure the performance improvement achieved by self-correction. For example, if a task requires that the model response should meet criteria that can be easily specified

in the initial instruction (e.g., the output should contain certain words, the generated code should be efficient, the sentiment should be positive, etc.), instead of including such requirements only in the feedback prompt, an appropriate comparison would be to directly and explicitly incorporate these requirements into the prompt for generating initial responses. Otherwise, when the instruction for generating initial predictions is not informative enough, even if the performance improves, it is unclear whether the improvement merely comes from more detailed instructions in the feedback prompt or from the self-correction step itself.

To illustrate such prompt design issues in the self-correction evaluation of some prior work, we take the Constrained Generation task in Madaan et al. (2023) as an example, where the task requires models to generate coherent sentences using all 20-30 input concepts. The original prompt in Madaan et al. (2023) (Figure 7) does not clearly specify that the LLM needs to include *all* concepts in the prompt; thus, they show that self-correction improves task performance by asking the model to identify missing concepts and then guiding it to incorporate these concepts through feedback.

Based on this observation, we add the following instruction "*Write a reasonable paragraph that includes \*ALL\* of the above concepts*" to the prompt for initial response generation (refer to Figure 8 for the full prompt). Following Madaan et al. (2023), we use concept coverage as the metric. We reference their results and replicate their experiments using `gpt-3.5-turbo-0613`. Table 8 demonstrates that our new prompt, denoted as *Standard Prompting (ours)*, significantly outperforms the results after self-correction of Madaan et al. (2023), and applying their self-correction prompt on top of model responses from our stronger version of the standard prompting again leads to a decrease in performance.

## 6   CONCLUSION AND DISCUSSION

Our work shows that current LLMs struggle to self-correct their reasoning without external feedback. This implies that expecting these models to inherently recognize and rectify their reasoning mistakes is overly optimistic so far. In light of these findings, it is imperative for the community to approach the concept of self-correction with a discerning perspective, acknowledging its potential and recognizing its boundaries. By doing so, we can better equip the self-correction technique to address the limitations of LLMs and develop the next generation of LLMs with enhanced capabilities. In the following, we provide insights into scenarios where self-correction shows the potential strengths and offer guidelines on the experimental design of future self-correction techniques to ensure a fair comparison.

**Leveraging external feedback for correction.** In this work, we demonstrate that current LLMs cannot improve their reasoning performance through intrinsic self-correction. Therefore, when valid external feedback is available, it is beneficial to leverage it properly to enhance model performance. For example, Chen et al. (2023b) show that LLMs can significantly improve their code generation performance through self-debugging by including code execution results in the feedback prompt to fix issues in the predicted code. In particular, when the problem description clearly specifies the intended code execution behavior, e.g., with unit tests, the code executor serves as the perfect verifier to judge the correctness of predicted programs, while the error messages also provide informative feedback that guides the LLMs to improve their responses. Gou et al. (2023) demonstrate that LLMs can more effectively verify and correct their responses when interacting with various external tools such as search engines and calculators. Cobbe et al. (2021); Lightman et al. (2023); Wang et al. (2023b) train a verifier or a critique model on a high-quality dataset to verify or refine LLM outputs, which can be used to provide feedback for correcting prediction errors. Besides automatically generated external feedback, we also often provide feedback ourselves when interacting with LLMs, guiding them to produce the content we desire. Designing techniques that enable LLMs to interact with the external environment and learn from different kinds of available feedback is a promising direction for future work.

**Evaluating self-correction against baselines with comparable inference costs.** By design, self-correction requires additional LLM calls, thereby increasing the costs for encoding and generating extra tokens. Section 4 demonstrates that the performance of asking the LLM to produce a final response based on multiple previous responses, such as with the multi-agent debate approach, is inferior to that of self-consistency (Wang et al., 2022) with the same number of responses. Regarding

this, we encourage future work proposing new self-correction methods to always include an in-depth inference cost analysis to substantiate claims of performance improvement. Moreover, strong baselines that leverage multiple model responses, like self-consistency, should be used for comparison. An implication for future work is to develop models with a higher probability of decoding the optimal solution in their answer distributions, possibly through some alignment techniques. This would enable the model to generate better responses without necessitating multiple generations.

**Putting equal efforts into prompt design.** As discussed in Section 5, to gain a better understanding of the improvements achieved by self-correction, it is important to include a complete task description in the prompt for generating initial responses, rather than leaving part of the task description for the feedback prompt. Broadly speaking, equal effort should be invested in designing the prompts for initial response generation and for self-correction; otherwise, the results could be misleading.

## 7 LIMITATIONS AND BROADER IMPACT

Although we have conducted a comprehensive evaluation spanning a variety of self-correction strategies, prompts, and benchmarks, our work focuses on evaluating reasoning of LLMs. Thus, it is plausible that there exist self-correction strategies that could enhance LLM performance in other domains. For example, prior works have demonstrated the successful usage of self-correction that aligns model responses with specific preferences, such as altering the style of responses or enhancing their safety (Bai et al., 2022; Ganguli et al., 2023; Madaan et al., 2023). A key distinction arises in the capability of LLMs to accurately assess their responses in relation to the given tasks. For example, LLMs can properly evaluate whether a response is inappropriate (Ganguli et al., 2023), but they may struggle to identify errors in their reasoning.

Furthermore, several prior works have already shown that LLM self-correction performance becomes significantly weaker without access to external feedback (Gou et al., 2023; Zhou et al., 2023a) and can be easily biased by misleading feedback (Wang et al., 2023a), which is consistent with our findings in this work. However, we still identified prevailing ambiguity in the wider community. Some existing literature may inadvertently contribute to this confusion, either by relegating crucial details about label usage to less prominent sections or by failing to clarify that their designed self-correction strategies actually incorporate external feedback. Regarding this, our paper serves as a call to action, urging researchers to approach this domain with a discerning and critical perspective. We also encourage future research to explore approaches that can genuinely enhance reasoning.

## REPRODUCIBILITY STATEMENT

Our experiments utilize GPT-3.5 and GPT-4, which are accessible via the public API at `https://platform.openai.com/docs/models`, as well as Llama-2, an open-source model. To facilitate reproducibility, we detail the specific kernels used, e.g., `gpt-3.5-turbo-0613`, or provide the access times for each experiment. We use prompts from previous works when possible. For our designed prompts, we include the exact prompts in Appendix A.

## ACKNOWLEDGEMENT

We would like to thank Chen Liang, William Cohen, Uri Alon, and other colleagues at Google DeepMind for valuable discussion and feedback.

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

## A PROMPTS AND EXAMPLE OUTPUTS

---

**Can you solve the following math problem? Christina is planning a birthday party and needs .75 gift bags per invited guest, because 1/4 of attendees don't show up. She invited 16 friends. Gift bags are $2 each. How much will she spend? Explain your reasoning. Your final answer should be a single numerical number, in the form \boxed{answer}, at the end of your response.**

If Christina invited 16 friends, then 1/4 of them won't show up, which means there will be 16 * 1/4 = 4 friends who won't attend the party. To determine how many gift bags to buy, we need to subtract the number of friends who won't show up from the total number of invited guests. The number of invited guests is 16 - 4 = 12. Since Christina needs 0.75 gift bags per invited guest, we can multiply this quantity by the number of invited guests to find the total number of gift bags needed, which is 0.75 * 12 = 9 gift bags. If each gift bag is $2, then Christina will spend a total of 2 * 9 = $<<2*9=18>>18 on gift bags. Therefore, Christina will spend $18 on gift bags. Answer: \boxed{18}. (Incorrect)

**Review your previous answer and find problems with your answer.**

Upon reviewing my previous answer, I recognize a mistake. I incorrectly stated that Christina needs 0.75 gift bags per invited guest. The given information states that 1/4 of the attendees won't show up, so the correct calculation should take this into account. To calculate the number of gift bags needed, we will multiply the number of invited guests by the ratio of attendees who will actually show up (75%). The number of invited guests is 16, and if 1/4 of them won't show up, then the ratio of attendees who will show up is 1 - 1/4 = 3/4. Multiplying the number of invited guests by the ratio of attendees who will show up gives us: 16 * (3/4) = 12 gift bags needed. Since each gift bag costs $2, the total amount Christina will spend on gift bags is: 12 * $2 = $<<12*2=24>>24.

**Based on the problems you found, improve your answer. Please reiterate your answer, with your final answer a single numerical number, in the form \boxed{answer}.**

Upon reviewing my previous answer, I recognize a mistake. I incorrectly stated that Christina needs 0.75 gift bags per invited guest. The given information states that 1/4 of the attendees won't show up, so the correct calculation should take this into account. To calculate the number of gift bags needed, we will multiply the number of invited guests by the ratio of attendees who will actually show up (75%). The number of invited guests is 16, and if 1/4 of them won't show up, then the ratio of attendees who will show up is 1 - 1/4 = 3/4. Multiplying the number of invited guests by the ratio of attendees who will show up gives us: 16 * (3/4) = 12 gift bags needed. Since each gift bag costs $2, the total amount Christina will spend on gift bags is: 12 * $2 = $<<12*2=24>>24. Therefore, Christina will spend $24 on gift bags. Answer: \boxed{24}. (Correct)

---

Figure 3: Example on GSM8K where self-correction changes an incorrect answer to a correct one. Text in **bold** denotes user prompts.

---

**Can you solve the following math problem? Terry eats 2 yogurts a day. They are currently on sale at 4 yogurts for $5.00. How much does he spend on yogurt over 30 days? Explain your reasoning. Your final answer should be a single numerical number, in the form \boxed{answer}, at the end of your response.**

First, we need to calculate how many sets of 4 yogurts Terry buys in 30 days. Since Terry eats 2 yogurts a day, he will need 2/4 = 0.5 sets of 4 yogurts per day. Next, we multiply the number of sets by the price of each set to calculate how much Terry spends per day. Each set of 4 yogurts costs $5.00, so Terry spends 0.5 * $5.00 = $<<0.5*5=2.50>>2.50 on yogurt per day. Finally, we multiply the amount Terry spends per day by the number of days to find the total amount he spends on yogurt over 30 days. Terry spends $2.50 per day for 30 days, so he spends $2.50 * 30 = $<<2.5*30=75.00>>75.00 on yogurt over 30 days. So, Terry spends $\boxed{75.00}$ on yogurt over 30 days. (Correct)

**Review your previous answer and find problems with your answer.**

Upon reviewing my previous answer, I realized that I made a mistake in calculating Terry's spending on yogurt over 30 days. I incorrectly stated that Terry spends $2.50 per day for 30 days, resulting in a total of $75.00. However, since Terry eats 2 yogurts a day and the yogurts are sold in sets of 4 for $5.00, Terry would actually need to buy 1 set of 4 yogurts to last for 2 days. Therefore, his spending on yogurt per set would be $5.00 for 2 days, or $5.00/2 = $2.50 per day. To calculate the total amount spent on yogurt over 30 days, we would multiply the amount spent per day ($2.50) by the number of sets needed in 30 days (which is 30 divided by 2 days per set). Therefore, the correct calculation would be $2.50 * (30/2) = $2.50 * 15 = $37.50. Apologies for the mistake. The correct answer is Terry spends $\boxed{37.50}$ on yogurt over 30 days.

**Based on the problems you found, improve your answer. Please reiterate your answer, with your final answer a single numerical number, in the form \boxed{answer}.**

Apologies for the error in my previous response. Upon reviewing the problem, here is the correct answer: Since Terry eats 2 yogurts a day and the yogurts are sold in sets of 4 for $5.00, Terry needs to buy 1 set of 4 yogurts to last for 2 days. Therefore, Terry's spending on yogurt per set is $5.00 for 2 days, or $5.00/2 = $2.50 per day. To calculate the total amount spent on yogurt over 30 days, we multiply the amount spent per day ($2.50) by the number of sets needed in 30 days. In this case, Terry needs 30/2 = 15 sets of yogurts over 30 days. Multiplying $2.50 by 15, we find that Terry spends a total of $2.50 * 15 = $37.50 on yogurt over 30 days. Therefore, the final answer is Terry spends $\boxed{37.50}$ on yogurt over 30 days. (Incorrect)

---

Figure 4: Example on GSM8K where self-correction changes a correct answer to an incorrect one.

**Can you solve the following math problem? Toulouse has twice as many sheep as Charleston. Charleston has 4 times as many sheep as Seattle. How many sheep do Toulouse, Charleston, and Seattle have together if Seattle has 20 sheep? Explain your reasoning. Your final answer should be a single numerical number, in the form \boxed{answer}, at the end of your response.**

Let's start by figuring out how many sheep Charleston has. We are given that Charleston has 4 times as many sheep as Seattle, so we can calculate this by multiplying Seattle's number of sheep by 4: Charleston = 4 * 20 = 80 sheep. Next, we are told that Toulouse has twice as many sheep as Charleston. Therefore, we can find Toulouse's number of sheep by multiplying Charleston's number of sheep by 2: Toulouse = 2 * 80 = 160 sheep. Finally, to find the total number of sheep, we add up the number of sheep for each city: Total = Seattle + Charleston + Toulouse = 20 + 80 + 160 = 260 sheep. Therefore, Toulouse, Charleston, and Seattle have a total of 260 sheep. Answer: \boxed{260}. (Correct)

**Review your previous answer and find problems with your answer.**

I apologize for the mistake in my previous response. Let's go through the problem again and find the correct answer. We are given that Charleston has 4 times as many sheep as Seattle. If Seattle has 20 sheep, then Charleston would have: Charleston = 4 * 20 = 80 sheep. Next, we are told that Toulouse has twice as many sheep as Charleston. Therefore, Toulouse would have: Toulouse = 2 * 80 = 160 sheep. To find the total number of sheep, we add up the number of sheep for each city: Total = Seattle + Charleston + Toulouse = 20 + 80 + 160 = 260 sheep. After reviewing my previous answer, I have determined that there are no problems with my updated response. The correct answer is 260 sheep.

**Based on the problems you found, improve your answer. Please reiterate your answer, with your final answer a single numerical number, in the form \boxed{answer}.**

Apologies for the confusion. Let me provide an improved answer. We are given that Charleston has 4 times as many sheep as Seattle, and Seattle has 20 sheep. So, Charleston would have 4 * 20 = 80 sheep. According to the information, Toulouse has twice as many sheep as Charleston. Therefore, Toulouse would have 2 * 80 = 160 sheep. To determine the total number of sheep, we add up the number of sheep for each city: Total = Seattle + Charleston + Toulouse = 20 + 80 + 160 = 260 sheep. After reviewing the previous answer, the correct final answer is \boxed{260}. (Correct)

Figure 5: Example on GSM8K where self-correction does not change the answer.

---

**Q: A fencing thrust with a sharp sword towards a person would result in what?**
**Answer Choices: (A) injury (B) small cuts (C) fever (D) competition (E) puncture wound.**
**Explain your reasoning. You must choose only one option from A to E. Your final answer should be a single letter from A to E, in the form (answer), at the end of your response.**

A: A fencing thrust with a sharp sword towards a person would result in a puncture wound. This is because a fencing thrust involves forcefully stabbing with a sword, which can penetrate the skin and create a deep, narrow wound. Puncture wounds can be very dangerous as they can damage internal organs and cause bleeding. Therefore, the most appropriate answer is (E) puncture wound. (Correct)

**Review your previous answer and find problems with your answer.**

Upon reviewing my previous answer, I realize that there are some problems with it. While a fencing thrust with a sharp sword can potentially result in a puncture wound, it is important to note that in a controlled fencing match, the swords used are typically blunted or have safety tips, designed to minimize the risk of injury. Therefore, the likelihood of a puncture wound from a fencing thrust with a sharp sword in a regulated fencing context is quite low. A more appropriate and accurate answer would be (D) competition, as a fencing thrust with a sharp sword is typically performed in the context of a fencing match or competition, rather than with the intention to cause injury.

**Based on the problems you found, improve your answer. You must choose only one option from A to E. Please reiterate your answer, with your final answer a single letter from A to E, in the form (answer).**

Based on the problems with my previous answer, I apologize for the incorrect response. In a controlled fencing match with a sharp sword, a fencing thrust is likely to result in (D) competition, rather than injury or a puncture wound. Therefore, the correct final answer is (D) competition. (Incorrect)

---

Figure 6: Example on CommonSenseQA where self-correction changes a correct answer to an incorrect one.

---

```
Concepts: ['footage', 'motion', 'ruin', 'tilt', 'window']

Sentence: time lapse footage with tilt up motion of the sun streaking
through window of ruin

###

Concepts: ['cause', 'hate', 'hut', 'local', 'love']

Sentence: new beach huts on the island have caused some controversy some
locals love them others hate them

[8 more in-context examples]

###

Concepts:  ['vest', 'snow','remove', 'silhouette', 'dunk', 'step',
'item', 'toddler', 'plaster', 'wall', 'piece', 'tool', 'circle',
'pull','wrap', 'headphone', 'deck', 'suit', 'press', 'mother', 'pit',
'cupcake', 'priest', 'rink', 'counter', 'rest','piano', 'wine', 'serve',
'winter']

Sentence: A toddler wearing a vest and snow boots steps onto the icy
deck, removes a circle-shaped piece of plaster from the wall, and pulls
out a tool from their winter suit to wrap their head around a headphone.
```

---

Figure 7: Example output for Constrained Generation using the prompt from Madaan et al. (2023). The constraint that the output needs to contain all the concepts is not explicitly mentioned in the prompt for initial response generation. However, in the feedback & refine prompts, the constraint is implied by "what concepts from the concept list are missing from the sentence?" (refer to Figures 25 and 26 of Madaan et al. (2023) for the feedback & refine prompts).

---

```
Concepts:  ['vest', 'snow','remove', 'silhouette', 'dunk', 'step',
'item', 'toddler', 'plaster', 'wall', 'piece', 'tool', 'circle',
'pull','wrap', 'headphone', 'deck', 'suit', 'press', 'mother', 'pit',
'cupcake', 'priest', 'rink', 'counter', 'rest','piano', 'wine', 'serve',
'winter']

Write a reasonable paragraph that includes *ALL* of the above concepts.

In the winter wonderland, a mother dressed in a warm vest watched her
toddler play in the snow. As the child attempted to remove the snow from
the ground, their silhouette created an enchanting image against the
white backdrop. Nearby, a group of friends gathered around a basketball
hoop, taking turns to dunk the ball with skillful steps. One of them,
holding an item resembling a tool, used it to plaster a piece of the wall
that had been damaged. Meanwhile, on the other side of the park, a circle
of children pulled their hats tightly to wrap around their heads,
protecting their ears from the cold wind as they listened to music
through their colorful headphones. On the deck of a nearby house, a man
in a sleek suit pressed a button, causing a hidden mechanism to reveal a
grand piano. As the music filled the air, the host served glasses of wine
to the guests, creating a warm and inviting atmosphere in the winter
evening. In the distance, a priest skated gracefully in an ice rink, his
movements mirroring the smoothness of his words during a sermon. At the
counter of a bakery, a customer ordered a cupcake and watched as the
baker skillfully decorated it with a winter-themed design. Not far away,
a group of friends gathered around a pit, roasting marshmallows and
enjoying the warmth of the fire. The scene painted a picture of joy,
friendship, and the beauty of winter.
```

---

Figure 8: Example output for Constrained Generation using a prompt casually written by the authors.

