# OpenReview forum: "Large Language Models Cannot Self-Correct Reasoning Yet"
_ICLR.cc/2024/Conference — ICLR 2024 poster_

### Official Review · Reviewer_oT3c · 2023-10-29

**Soundness:** 3 good
**Presentation:** 3 good
**Contribution:** 4 excellent
**Rating:** 8
**Confidence:** 4

**Summary:**

Large Language Models (LLMs) have been increasingly capable. However, they still make many mistakes. Recent work has explored the idea of “self-correction” where LLMs refine their responses based on feedback to their previous outputs. This paper critically examines the role and efficacy of self-correction within LLMs. Central to the investigation is the definition of intrinsic self-correction, whereby an LLM attempts to correct its initial responses based solely on its inherent capabilities, with no external feedback. The paper finds that LLMs struggle to self-correct their responses for reasoning tasks without external feedback, and at times, their performance might even degrade post self-correction.

**Strengths:**

1. Contrary to prior results, the paper finds the self-correct methods in prior research such as Kim et al. (2023); Shinn et al. (2023) make use of oracle labels to guide the self-correction process.

2. For self-correction through multi-agent debate (Du et al., 2023; Liang et al., 2023) to improve reasoning where multiple instances of an LLM critique each other’s responses, the paper's results reveal that its efficacy is no better than self-consistency when considering an equivalent number of responses, highlighting the limitations of such an approach.

**Weaknesses:**

1. Intrinsic self-correction, defined as the model endeavors to rectify its initial responses based solely on its inherent capabilities without the crutch of external feedback, is not very clear to me. Does recall examples from parameter knowledge (see paper below) considered intrinsic self-correction?
Large Language Models as Analogical Reasoners
https://arxiv.org/pdf/2310.01714

2. It is not clear which prior papers have the various problems exposed. It would be very helpful to put them in a table. Furthermore, please provide details on where prior methods rely on oracle labels, specific problems on poorly constructed pre-prompts, specific benchmark results that are wrong. For example, I do not seem to locate which part Shinn et al. (2023) have the problems.

**Questions:**

The paper exposes an intriguing problem in prior work on self-correction of LLMs that shows, to the contrary that LLMs can not self-correct reasoning yet.

However, it is still not clear how we should think about all the techniques to improve reasoning without external feedback. Does breakdown a problem into sub-problem and do step-wise verification paired with self-consistency considered self-correction? It would be very helpful to put all these techniques into perspective.

---

> ### Author Response · Authors · 2023-11-17
> **Official Response to Reviewer oT3c**
>
> Thank you for your positive review and the valuable points raised. We would like to address each of your questions.
>
> >**1. Intrinsic Self-Correction and Analogical Prompting:**
>
> Thank you for your question. Analogical prompting, which involves recalling relevant problems and solutions from parameter knowledge, is a form of pre-hoc prompting. In contrast, self-correction represents a type of post-hoc prompting (see Section 4 for more details). These methods differ fundamentally; hence, analogical prompting is not directly related to self-correction.
>
>
> >**2. “It is not clear which prior papers have the various problems exposed…”**
>
> Thank you for your suggestion. The table below summarizes the issues of prior methods as you suggested, and we added this table to Appendix C in our revision.
>
> | Method                                              | Issue                                      |
> |-----------------------------------------------------|--------------------------------------------|
> | RCI (Kim et al., 2023); Reflexion (Shinn et al., 2023) | Use of oracle labels                        |
> | Multi-Agent Debate (Du et al., 2023)                | Unfair comparison to self-consistency       |
> | Self-Refine (Madaan et al., 2023)                   | Suboptimal pre-hoc prompt design            |
>
> Regarding the use of oracle labels, in Appendix C.2 of Kim et al. (2023), they stated that “we use the correct label to decide when to stop the RCI loop.” For Shinn et al. (2023), please refer to Section 4.2 of their paper: “we use exact match answer grading using the environment to give a binary success signal to the agent,” which is also revealed in their official implementation.
>
> >**3. Question: “However, it is still not clear how we should think about all the techniques to improve reasoning without external feedback…”**
>
> Thank you for your suggestion. As discussed in our Section 5 “Employing self-consistency as a method of self-verification”, breaking down a problem into sub-problems and conducting step-wise verification paired with self-consistency can be helpful.
>
> Another promising future direction is exploring how to enable LLMs to interact with and learn from external feedback (e.g., interacting with the environment), though this falls outside the scope of "intrinsic self-correction" discussed in our paper.
>
> We thank you for recognizing the contribution of our paper to the field. Your feedback will significantly enhance the clarity and comprehensiveness of our research.

---

### Official Review · Reviewer_N1pE · 2023-10-31

**Soundness:** 3 good
**Presentation:** 4 excellent
**Contribution:** 3 good
**Rating:** 8
**Confidence:** 4

**Summary:**

The paper studies *intrinsic* self-correctness in LLMs in the context of reasoning, where no external feedback is provided to the language model i.e., just simply asking the model to detect a mistake in its output and fix it. Through experiments over three reasoning tasks (GSM8K, Commonsense QA, and HotpotQA), the paper does the following. First, it argues that what is currently referred to in the literature as "self-correctness" where external feedback is provided (e.g., whether the final answer is correct) is not practical and we should focus on settings where we do not know the answer. Second, self-consistency is a strong baseline and similar approaches such as multi-agent debate methods should be considered an instance of voting rather than self-correction methods. Third, to assess whether LLMs actually have the capacity for self-correction, the authors emphasize the importance of designing a good initial (pre-hoc) prompt that performs well as opposed to using a suboptimal initial prompt, where providing additional information in the feedback prompt can be useful.  Overall, the paper argues that current LLMs fall short when prompted to self-correct their reasoning and when no additional information is provided in the feedback prompt.

**Strengths:**

* The paper studies an important direction that is now taking over the LLM scene and brings a fresh perspective on how good SoTA LLMs are at detecting their own errors.
* Focusing on *intrinsic self-correction* is much needed in the current "sea" of self-correction papers.
* The experimental design is sound. I liked the random guessing baseline with Commonsense QA.
* I have to say I enjoyed reading the paper: the flow is natural, the writing is good, and most of the arguments are intuitive and make sense.

Overall the community would certainly benefit from this paper gaining wider visibility.

**Weaknesses:**

* I find the explanation in section 3.2.1—why post-hoc prompting can lead the model to go from a correct to an incorrect answer—unsatisfying. We know that the feedback prompt is changing the model output somehow. The question is *why?* I suggest providing more intuition here.
* The paper discusses the issue but does not provide any hint at a potential solution. I understand this is not the point of the paper but hinting at potential directions to improve intrinsic self-correctness could make the paper even more valuable.
* The authors focus only on ChatGPT and GPT-4. I think the community could benefit from seeing that the results discussed generalize to other open-source LLMs such as LLaMA

**Questions:**

* Have you tried combining Multi-agent debating with self-consistency?
* How much effort did you invest into finding the self-correct prompt you used "Review your previous answer and find problems with your answer"? Can't different variations of this same prompt lead to different results?

---

> ### Author Response · Authors · 2023-11-17
> **Official Response to Reviewer N1pE**
>
> We thank the reviewer for highlighting the importance and soundness of our study, as well as the presentation of our paper. We greatly appreciate your thorough review and constructive feedback. We are pleased to address the points you raised.
>
> >**1. Explanation of Post-Hoc Prompting Leading to Incorrect Answers:**
>
> Thank you for your feedback. The fundamental issue is that LLMs cannot properly judge the correctness of their reasoning. So the feedback prompt introduces a new context that may bias the model away from its original correct answer. We have added the explanation in the revision.
>
> >**2. Potential Directions for Improving Intrinsic Self-Correction:**
>
> Thank you for your suggestion. We have provided some suggestions for future research and practical application in Section 5. For intrinsic self-correction, it can still be useful for aligning responses with certain preferences. Regarding self-correcting reasoning, one suggestion involves leveraging self-consistency for self-verification. Additionally, exploring ways to enable LLMs to interact with and learn from external feedback (e.g., interacting with the environment) would be a potential solution and a promising future direction.
>
> >**3. Generalization to Other LLMs:**
>
> Thank you for your suggestion. We conducted additional experiments with GPT-4-Turbo (gpt-4-1106-preview in the table) and Llama-2-70B. The results consistently show that the models’ performance decreases after self-correction. We added the results and full error analysis in Appendix B.2 of our revision.
>
> |                          |                      | # calls | GSM8K | CommonSenseQA |
> |--------------------------|----------------------|-------|-------|---------------|
> | gpt-4-1106-preview       | Standard Prompting   | 1     | 91.5  | 84.0          |
> |                          | Self-Correct (round 1) | 3   | 88.0  | 81.5          |
> |                          | Self-Correct (round 2) | 5   | 90.0  | 83.0          |
> | Llama-2-70b-chat-hf      | Standard Prompting   | 1     | 62.0  | 64.0          |
> |                          | Self-Correct (round 1) | 3   | 43.5  | 37.5          |
> |                          | Self-Correct (round 2) | 5   | 36.5  | 36.5          |
>
> >**4. Question A: Combining Multi-Agent Debating with Self-Consistency**
>
> Yes. When we run the multi-agent debate method, we take the majority vote of the final responses of all three agents (following the setting of the original paper). We also try taking the majority vote of all responses (including the previous responses) of the agents, but do not observe performance improvement.
>
>
> >**5. Question B: Effort Invested in Finding the Self-Correct Prompt:**
>
> The prompt "Review your previous answer and find problems with your answer" originates from Kim et al., (2023). We adhered to their prompt to replicate their experiments. We have also made efforts in designing different prompts for testing:
> - In Appendix B.1 of the initial draft, we test different feedback prompts, such as “Verify whether your answer is correct, and provide an explanation.”
> - In Appendix B.2 of the revised draft, we add a prompt suggested by other reviewers.
> - The prompts used in previous works (Kim et al., 2023; Shinn et al., 2023; Du et al., 2023; Madaan et al., 2023) are different.We adhere to the prompts from the source papers to replicate their experiments. So our prompts in these experiments are also different.
>
> All the results consistently show that the models’ performance on reasoning does not improve and even decreases after self-correction.

---

> > ### Comment · Reviewer_N1pE · 2023-11-21
> >
> > I thank the authors for the additional experiments and results. I have decided to raise my score from 6 to 8.

---

> > > ### Author Response · Authors · 2023-11-21
> > >
> > > Thank you. We are happy that you like our experiments, and we thank you again for your insightful comments.

---

### Official Review · Reviewer_No3p · 2023-11-02

**Soundness:** 3 good
**Presentation:** 4 excellent
**Contribution:** 3 good
**Rating:** 6
**Confidence:** 4

**Summary:**

This work investigates prior reports that LLMs can self-correct their own responses when prompted to do so. A critical distinction is made between self-correction with and without external feedback (termed intrinsic self-correction). The results show that given the same self-correction prompt used in prior work on particular benchmark datasets, LLMs often do not succeed in self-correcting their responses without external feedback.

**Strengths:**

Self-correction of today’s LLMs is a highly significant topic. The paper clearly points out the crucial distinction between self-correction with and without feedback, and sheds light on the latter case (intrinsic self-correction). The usage of oracle labels to terminate self-critique is also examined.

The paper’s organization and writing quality is uniformly high, making it a pleasure to read.

**Weaknesses:**

The most serious weakness of the paper is its misleading title, which baldly asserts a claim unsupported by the analysis and results. The words “cannot” and “yet” imply that even today’s most capable LLMs (GPT-4) obtain zero benefit from self-correction in nearly all cases. The abstract quickly tones down the claim by saying “our research indicates that LLMs struggle to self-correct their responses without external feedback”, but even that statement goes beyond what is actually demonstrated by the experiments. A more properly measured title for this work would be “Reexamining the Ability of Large Language Models to Self-Correct”. This is more like statements that appear later in the paper:  “we provide insights into the nuances of LLMs’ self-correction capabilities”, and “their self-correction capabilities, particularly in reasoning, are still nascent.”

The second major problem is the work’s heavy reliance on a single, loaded prompt:
“Review your previous answer and find problems with your answer”

Practitioners in the rapidly moving field of prompt engineering recognize this as a highly leading prompt, essentially telling the LLM that problems do exist in the previous answer. This typically causes the LLM to find problems that aren’t actually present. As the paper says at one point:  “careful consideration of prompt design is essential”.

Here’s a longer list of important factors used routinely in prompt engineering that would be needed for a proper study of LLM self-correction:

- Focus on GPT-4, since its capabilities are known to be significantly greater than those of GPT-3.5. This is reflected in Table 3.

- Include diagrams like those in Figure 1 for GPT-4, not just for GPT-3.5.

- Evaluate a set of reasonable, unbiased self-correction prompts. For instance: “Assume that this answer could be either correct or incorrect. Review the answer carefully and report any serious problems you find.”

- Focus on the zero-temperature setting, since that’s far less prone to spurious hallucination than 1.

In discussing the option of trying other self-correction prompts (“Such a search essentially leverages feedback from humans or training examples.”), the paper conflates feedback received on a per-problem basis (as when using an oracle), with feedback received from the results of multiple self-correction prompts across entire datasets. The former does indeed go beyond the definition of intrinsic self-correction, but the latter is merely hard work.

The paper says that “Our main objective is to encourage a more critical examination of self-correction experiments.” But readers expect this paper to be a critical examination of that nature, not just a call for critical examination.

The paper also states that “in the reasoning tasks studied in this paper, we did not observe any improvement through self-correction.” That’s true enough, but as pointed out above, the study was not carried far enough to shed much light on the general question of how reliably LLMs can self-correct.

**Post-rebuttal Comments**

I commend the authors for performing the additional experiments reported in Appendix B.2, using GPT-4 with an unbiased feedback prompt and zero temperature. These results on these two datasets are interesting, so I have raised my assessment of the paper’s contribution from 2 to 3.

I still view these limited results as insufficient to support the broad claim made by the paper’s title:  “Large Language Models Cannot Self-Correct Reasoning Yet”. The authors argue that the title’s claim is restricted to “reasoning” tasks, but this is not much of a restriction at all.

Regarding other feedback prompts used in the experiments, the ones in Appendix B.1 only apply to GPT-3.5, which is widely known to not be good at self-critique. The authors imply that the feedback prompts from prior works are different, but Appendix A shows only the single, loaded prompt: “Review your previous answer and find problems with your answer”

For these reasons, I still view the work as fundamentally unsound, in the sense that the limited findings do not justify the headline-grabbing title of the paper.

**Additional Post-rebuttal Comments**


*we do not fully understand the argument that "The authors argue that the title’s claim is restricted to “reasoning” tasks, but this is not much of a restriction at all."*


LLMs are strong at memorization, but struggle with many kinds of reasoning. So while self-critique can be applied to both memorization and reasoning problems, application to reasoning is of greater interest and is being intensely studied. For this reason, restricting the consideration of self-critique to reasoning is not much of a restriction at all.



*we will change the title to **"Reexamining the Ability of Large Language Models to Self-Correct Reasoning"** in the final version.*


This title would be in line with the experiments. So on the assumption that this will indeed be the title, I'm raising my rating from 3 to 6.

**Questions:**

Section 3.2 says “Intuitive Explanation. If the model is well-aligned and paired with a thoughtfully designed initial prompt, the initial response should already be optimal”  But why should this be intuitive or expected? Wouldn’t similar reasoning conclude that the value of chain-of-thought prompting is itself unintuitive?

---

> ### Author Response · Authors · 2023-11-17
> **Official Response to Reviewer No3p [1/2]**
>
> We thank the reviewer for highlighting the importance of our study and commending the quality of our writing. We appreciate the insightful and detailed feedback provided. We would like to address each of your concerns.
>
> >**1. Title and Claim Adjustment:**
>
> Thank you for your feedback. Based on the ICLR policy, we are not able to change the paper title during the rebuttal phase, but we will consider your suggestion in the final version. Nevertheless, we wish to clarify that our title does not mean self-correction does not work for all tasks; rather, as our original title includes the word “reasoning”, our work focuses on self-correction for reasoning. The title reflects our finding that current LLMs cannot yet enhance reasoning performance through self-correction. We have also adjusted our claims in the paper to make them more specific to reasoning.
>
> >**2. Test on Various Prompts:**
>
> A primary question raised by the reviewer concerns the reliance on a single, loaded prompt. However, this is not true. **In our paper, both in the main text and Appendix B, we have tested multiple prompts**.
>
> First, as this paper functions partially as a critique paper, we initially follow the prompts and models used in prior studies, specifically testing three different sets of prompts from Kim et al., 2023, Shinn et al., 2023, and Du et al., 2023.
>
> **Furthermore, we also test unbiased self-correction prompts in Appendix B.1 of the initial version of our paper, such as 'Verify whether your answer is correct, and provide an explanation.' In Appendix B.2 of the revised manuscript, we conduct additional tests on GPT-4-Turbo and Llama-2-70b using the prompt suggested by the reviewer: 'Assume that this answer could be either correct or incorrect. Review the answer carefully and report any serious problems you find.' The results consistently show that the models’ performance decreases after self-correction.** We welcome the reviewer to check the results.
>
>
> >**3. Use of Zero-Temperature Setting:**
>
> As our paper partially serves as a critique paper, our choice of temperature setting adheres to previous works. We conducted additional experiments detailed in Appendix B.2, employing the settings suggested by the reviewer (e.g., zero-temperature, the prompt suggested by the reviewer). We added the results and full error analysis in Appendix B.2 of our revision.
>
> |                          |                      | # calls | GSM8K | CommonSenseQA |
> |--------------------------|----------------------|-------|-------|---------------|
> | gpt-4-1106-preview       | Standard Prompting   | 1     | 91.5  | 84.0          |
> |                          | Self-Correct (round 1) | 3   | 88.0  | 81.5          |
> |                          | Self-Correct (round 2) | 5   | 90.0  | 83.0          |
> | Llama-2-70b-chat-hf      | Standard Prompting   | 1     | 62.0  | 64.0          |
> |                          | Self-Correct (round 1) | 3   | 43.5  | 37.5          |
> |                          | Self-Correct (round 2) | 5   | 36.5  | 36.5          |
>
>
> >**4. “Focus on GPT-4. Include diagrams like those in Figure 1 for GPT-4, not just for GPT-3.5.”**
>
> There are two primary reasons we test on GPT-3.5 (while also providing results for GPT-4):
>
> (1) Alignment with previous studies: As our paper partially serves as a critique of the studies by Kim et al., 2023; Shinn et al., 2023; and Du et al., 2023, which focus on GPT-3.5, it is imperative that we follow their settings to accurately reproduce their results.
>
> (2) Cost considerations: The cost associated with GPT-4 is substantial. For instance, conducting a single full test on GSM8K with self-correction using GPT-4 incurs a cost of approximately $200 (in total, it will cost thousands of dollars). To manage these costs, we opt to randomly sample 200 examples for our tests in our initial draft.
>
> We appreciate your suggestion and have included diagrams for GPT-4 in both Figure 1 and Appendix B.2.

---

> ### Author Response · Authors · 2023-11-17
> **Official Response to Reviewer No3p [2/2]**
>
> >**5. Our study focuses on the question "Can LLMs self-correct *reasoning* intrinsically?" by examining various prompts and previous studies, not on the general question of how reliably LLMs can self-correct for all tasks:**
>
> As discussed above, in Appendix B, we have tested with different (unbiased) self-correction prompts. We have also tested three different sets of prompts from Kim et al., 2023; Shinn et al., 2023; and Du et al., 2023. All the results consistently show that the models’ performance on reasoning does not improve and even decreases after self-correction, supporting the statement that LLMs struggle to self-correct reasoning intrinsically.
>
> Moreover, as mentioned in the title and our response 1 above, the focus of our paper is on self-correction for *reasoning*, not on general self-correction for all tasks. While we do discuss self-correction for other tasks a bit, our aim is to provide readers with a more balanced and comprehensive understanding of self-correction. Additionally, we are open to conducting any further experiments (considering the cost and time) that the reviewers might deem necessary before the revision deadline.
>
>
> >**6. Questions on Section 3.2:**
>
> The reviewer appears to have truncated our statement into an incomplete sentence. The complete sentence is, “If the model is well-aligned and paired with a thoughtfully designed initial prompt, the initial response should already be optimal *relative to the prompt and the specific decoding algorithm*.” The critical missing part, 'the response is optimal relative to the given prompt and the specific decoding algorithm', significantly alters the meaning when omitted.
>
> The effectiveness of 'Chain-of-Thought Prompting' lies in its provision of better guidance for the model to solve the problem. Therefore, the model's response is optimal to the improved prompt (w/ CoT); it surpasses the response generated without CoT, which is optimal to a less effective prompt (w/o CoT). This contrasts significantly with the intrinsic self-correction we discuss here, where the feedback prompt (without usage of oracle labels) does not provide any helpful benefits for reasoning and may even bias the model to generate a worse response.
>
> We thank you for your constructive critique and guidance. The suggested modifications certainly enhance our paper.

---

> ### Author Response · Authors · 2023-11-20
> **Follow-up Response to Reviewer No3p**
>
> Thank you for reviewing our response. We would like to follow up and further address your concerns.
>
> First, we present additional results for GPT-4-Turbo using different feedback prompts. These results are summarized in the table below, consistently demonstrating a decline in performance after self-correction. We have included these results in Table 8 of Appendix B.2.
>
> |                               | # calls | GSM8K | CommonSenseQA |
> |-------------------------------|---------|-------|---------------|
> | _Feedback Prompt_: Assume that this answer could be either correct or incorrect. Review the answer carefully and report any serious problems you find. |         |       |               |
> | Standard Prompting            | 1       | 91.5  | 84.0          |
> | Self-Correct (round 1)        | 3       | 88.0  | 81.5          |
> | Self-Correct (round 2)        | 5       | 90.0  | 83.0          |
> |                                   |         |       |               |
> | _Feedback Prompt_: Review your previous answer and determine whether it's correct. If wrong, find the problems with your answer. |         |       |               |
> | Standard Prompting            | 1       | 91.5  | 84.0          |
> | Self-Correct (round 1)        | 3       | 90.0  | 74.5          |
> | Self-Correct (round 2)        | 5       | 90.0  | 81.0          |
> |                                   |         |       |               |
> | _Feedback Prompt_: Verify whether your answer is correct, and provide an explanation. |         |       |               |
> | Standard Prompting            | 1       | 91.5  | 84.0          |
> | Self-Correct (round 1)        | 3       | 91.0  | 81.5          |
> | Self-Correct (round 2)        | 5       | 91.0  | 83.5          |
>
>
> Regarding the feedback prompts from previous studies: In Section 3.1.1 Prompts, we state, "we mostly adhere to the prompts from the source papers. For GSM8K and CommonSenseQA, we integrate format instructions into the prompts of Kim et al. (2023) to facilitate a more precise automatic evaluation (detailed prompts can be found in Appendix A). For HotpotQA, we use the same prompt as Shinn et al. (2023)." In Section 3.3, we note, "For an unbiased implementation, we use the exact same prompt as Du et al. (2023) and replicate their experiment with the gpt-3.5-turbo-0301 model, incorporating 3 agents and 2 rounds of debate." The reviewer may refer to the previous papers or their implementations for the prompts, which are identical to those used in our experiments.
>
> Regarding the title, we thank you for your suggestion. However, we do not fully understand the argument that "The authors argue that the title’s claim is restricted to “reasoning” tasks, but this is not much of a restriction at all." We would appreciate it if the reviewer could further clarify. If you feel that this is the main weakness, we will change the title to "Reexamining the Ability of Large Language Models to Self-Correct Reasoning" in the final version. Additionally, we have softened the statements in the paper, for example, changing 'cannot' to 'struggle to'.
>
> We hope the additional results and the responses above address your concerns. Please do not hesitate to contact us for any further questions or clarifications. Additionally, we are open to conducting any further experiments (considering the cost and time) that the reviewers might deem necessary before the revision deadline.

---

> ### Author Response · Authors · 2023-11-21
>
> Thank you for your update. We are happy that our response addressed your concern, and we thank you again for your constructive feedback.

---

### Official Review · Reviewer_yEtd · 2023-11-08

**Soundness:** 3 good
**Presentation:** 1 poor
**Contribution:** 2 fair
**Rating:** 5
**Confidence:** 3

**Summary:**

LLMs apparently have the ability to self-correct, as evidenced by previous publications. In the current article, which acts partly as a survey, the authors investigate how well LLMs can actually self-correct intrinsically and report nuanced, mixed results in terms of LLMs ability to perform such a feat.

**Strengths:**

The paper tackles a very important topic, has a good literature review section, and uses well-known and trusted datasets to investigate self-correction abilities. In particular, I appreciated the distinction between intrinsic self-correction and self-correction that leverages information from humans or training examples.

**Weaknesses:**

- Only a small set of questions (200) is used on GPT4, the remaining ones apply only to ChatGPT
- In terms of reasoning, there are much more challenging datasets out there
- I found the presentation somewhat confusing since there wasn't a clear description of their methodology (e.g., were all self-correction prompts formulated as the examples in Figure 2, or did variations exist?).
- Also, the wording is unfortunately often trendy rather than clear: From the conclusion: "while LLMs represent a groundbreaking step forward in the realm of AI and language generation, their self-correction capabilities, particularly in reasoning, are still nascent". "Nascent" can mean anything from the self-correction capabilities being "not present", "weak" "promising" etc. Please try to use more specific wording.
- The paper is confusing to read, because there are somewhat contradictory statements: In the beginning the authors emphasize that they want to investigate *intrinsic* self-correction abilities. Then, on page 3, they state: "With this in mind, we center our investigation on a pivotal query: Can large language models self-correct their reasoning?" which seems to me to not imply intrinsic self-correction, but any type of self-correction. This unclarity seems to pervade the remainder to the paper.
E.g., in section 7 the authors state " Some existing literature may inadvertently contribute to this confusion, either by
relegating crucial details about label usage to less prominent sections or by failing to clarify that their
designed self-correction strategies actually incorporate external feedback. Our intention in this paper is to amplify these concerns and offer a comprehensive overview of the state of “self-correction” in LLMs".
This implies that the current paper was more like a survey, rather than establishing new results.
Furthermore, the title suggests that self-correction does not work, although the paper actually does not really argue this and in section 7 the authors remark "The title, “Large Language Models Cannot Self-Correct Reasoning Yet”, is not an outright dismissal of self-correction techniques". Generally, it would be good to use a title that is completely representative of the paper. (Perhaps add "intrinsic" to the title?)

**Questions:**

- I don't understand the second paragraph from section "3.1.3 REFLECTION". How does this guessing approach work? It seems that cannot be used for non-multiple-choice type answers, like in the case of the GSM8K dataset?
I cannot follow your footnote: "For GSM8K, a similar random baseline might not exist, but the underlying rationale remains the same. Additionally, we can design a baseline, for example, by generating a random number each time. After a significant number of rounds, it may reach the correct answer, but such a kind of improvement is apparently not meaningful. A more direct justification is: If we already know the answer, why do we need to do this?"
- "Per the discussions in Section 3.1.3, since the idea that LLMs can self-correct their reasoning is not supported by the evidence so far, we turn our focus to the results in the intrinsic self-correction." I wasn't quite able to follow why section 3.1.3 shows this; Table 1 actually seems to show that self-reasoning is well-supported by evidence?

---

> ### Author Response · Authors · 2023-11-17
> **Official Response to Reviewer yEtd [1/2]**
>
> We thank the reviewer for recognizing the importance of our study. We appreciate your comprehensive review of our paper and would like to address each of your questions.
>
> >**1. Use of GPT-4 and Selection of Datasets**
>
> There are two primary reasons we test on GPT-3.5 (while also providing results for GPT-4):
>
> (1) Alignment with previous studies: As our paper partially serves as a critique of the studies by Kim et al., 2023; Shinn et al., 2023; and Du et al., 2023, which focus on GPT-3.5, it is imperative that we follow their settings (e.g., testing on GPT-3.5) to accurately reproduce their results on the well-established benchmarks used in their papers.
>
> (2) Cost considerations: The cost associated with GPT-4 is substantial. For instance, conducting a single full test on GSM8K with self-correction using GPT-4 incurs a cost of approximately $200 (in total, it will cost thousands of dollars). To manage the costs, we opt to randomly sample 200 examples for our tests on GPT-4.
>
> A point to mention is that we do consider running tests on the full test set when the cost is reasonable. For instance, Du et al. (2023) tested only 100 examples on GSM8K with GPT-3.5, whereas we chose to run the full test set to make the results more reliable.
>
> In Appendix B.2 of our revision, we conducted additional experiments on more examples with GPT-4-Turbo and Llama-2, employing the settings suggested by other reviewers.
>
> |                          |                      | # calls | GSM8K | CommonSenseQA |
> |--------------------------|----------------------|-------|-------|---------------|
> | gpt-4-1106-preview       | Standard Prompting   | 1     | 91.5  | 84.0          |
> |                          | Self-Correct (round 1) | 3   | 88.0  | 81.5          |
> |                          | Self-Correct (round 2) | 5   | 90.0  | 83.0          |
> | Llama-2-70b-chat-hf      | Standard Prompting   | 1     | 62.0  | 64.0          |
> |                          | Self-Correct (round 1) | 3   | 43.5  | 37.5          |
> |                          | Self-Correct (round 2) | 5   | 36.5  | 36.5          |
>
>
> >**2. “were all self-correction prompts formulated as the examples in Figure 2, or did variations exist?”**
>
> We test on various prompts:
> - In Section 3.1.1 Prompts, we mentioned that “we mostly adhere to the prompts from the source papers. For GSM8K and CommonSenseQA, we integrate format instructions into the prompts of Kim et al. (2023) to facilitate a more precise automatic evaluation (detailed prompts can be found in Appendix A). For HotpotQA, we use the same prompt as Shinn et al. (2023).” The prompts used in Du et al. (2023) and Madaan et al. (2023) are also different.
> - In Appendix B.1 of the initial draft, we test on different feedback prompts, such as “Verify whether your answer is correct, and provide an explanation.”
> - In Appendix B.2 of the revised draft, we add a prompt suggested by other reviewers.
>
> >**3. Presentation and Wording**
>
> Thank you for your feedback. Note that at the end of Section 2 (page 2), we specified, “For brevity, unless explicitly stated otherwise (e.g., self-correction with oracle feedback), all references to ‘self-correction’ in the remainder of this paper pertain to intrinsic self-correction.” Thus, the question “Can large language models self-correct their reasoning?” on page 3 refers to intrinsic self-correction.
>
> We appreciate your suggestion on the wording. The term “nascent” was intended to convey that LLMs' self-correction abilities, particularly in reasoning, are not yet fully effective. We have updated the conclusion section and incorporated more specific wording in the revision.
>
> >**4. “This implies that the current paper was more like a survey...”**
>
> We would like to clarify that our paper critiques and analyzes LLMs’ ability to self-correct reasoning. We present new results showing that LLM self-correction without oracle labels underperforms baselines without self-correction:
> - Critique: We reexamine previous works and find that 1) previous works use oracle labels to perform self-correction, which is not practical; 2) multi-agent debate is not superior to self-consistency; 3) there is an issue with employing a well-crafted post-hoc prompt to guide the model in “self-correcting” a response generated through a poorly constructed pre-hoc prompt.
> - Analysis: We demonstrate that current LLMs struggle to self-correct reasoning intrinsically through comprehensive analysis and offer insights and suggestions for future work.
>
> >**5. Paper Title**
>
> Thank you for your feedback. Based on the ICLR policy, we are not able to change the title during the rebuttal phase, but we will consider your suggestion in the final version. Nevertheless, we wish to clarify that our title doesn’t mean self-correction does not work for all tasks; rather, our work focuses on self-correction for reasoning. The title reflects our finding that current LLMs cannot yet enhance reasoning performance through intrinsic self-correction.

---

> ### Author Response · Authors · 2023-11-17
> **Official Response to Reviewer yEtd [2/2]**
>
> >**6. Question A: Explanation of random guessing baseline**
>
> This section presents a case study to demonstrate that improvements achieved using oracle labels (a setting used in previous work) do not necessarily reflect true self-correction ability. Specifically, we illustrate the effectiveness of a simple guessing strategy employing oracle labels in multiple-choice settings.
>
> For example, consider a question with the correct label (C). If the model initially predicts the answer as (A), we can determine the response's correctness using labels: since (A) ≠ (C), the random baseline is employed to randomly select a new answer from the remaining options (B-E). If the sampled option is (D), and (D) ≠ (C), the random baseline samples again from the remaining options (B, C, E). If the next sampled option is (C), which equals (C), we then obtain the correct answer and stop. In a 5-choice question format like CommonSenseQA, this baseline can always achieve 100% accuracy within four rounds of self-correction.
>
> For non-multiple-choice datasets like GSM8K (which can be regarded as questions with an extensive range of numerical choices), we mention the possibility of designing a baseline by generating random numbers in the footnote. Through a significant number of attempts, it may still reach the correct answer and achieve performance improvement. However, this type of improvement is not meaningful. In the revision, we remove the footnote to avoid confusion.
>
> >**7. Question B: “I wasn't quite able to follow why section 3.1.3 shows this; Table 1 actually seems to show that self-reasoning is well-supported by evidence?”**
>
> As explained above, the results in Table 1 use oracle labels to guide self-correction, which is impractical and differs from the intrinsic self-correction setup. Therefore, they cannot support the claim that LLMs can self-correct reasoning (intrinsically). We have adjusted the writing to enhance clarity.

---

> ### Author Response · Authors · 2023-11-21
>
> Dear Reviewer yEtd, we hope that our paper revision and the above responses address your concerns. Please let us know your thoughts, and we are more than happy to answer any further questions.

---

> > ### Comment · Reviewer_yEtd · 2023-11-23
> >
> > I commend the authors for improving their paper. Its readability has significantly increased. I found the cost aspect that the authors mentioned above striking - this would be an important, concrete piece of information to add to the paper that motivates the small dataset strongly.
> >
> > I believe the paper currently mentions somewhat vaguely only "costs"; I think adding concrete numbers (e.g., for the camera-ready version, if the paper is accepted), as they did above, relating to how much various runs cost would be 1) very interesting for the readers and 2) clearly show how this analysis could be scaled and what financial resources that would take, which would inform readers how easy/hard it is in financial terms to replicate these results.
> > Few papers do that, so adding this information could be a selling point of this paper.
> >
> > Regarding the score, I will maintain it.

---

> ### Public Comment · ~Baohao_Liao1 · 2024-07-16
> **Really ridiculous reviews**
>
> Such reviews drive PhDs away from research after their graduation.

---

### Author Response · Authors · 2023-11-17
**General Response to Reviewers**

We thank all reviewers for the constructive comments. We appreciate the reviewers' recognition of the importance and soundness of our study. We are also pleased that the reviewers found this paper a pleasure to read.

**1. Highlighting results of various prompts**

A primary question raised by the reviewers concerns prompt selection. In fact, we have tested various prompts:

- In Appendix B.1 of the initial draft, we test different feedback prompts, such as “Verify whether your answer is correct, and provide an explanation.”
- In Appendix B.2 of the revised draft, we add a prompt suggested by the reviewers.
- The prompts used in previous works (Kim et al., 2023; Shinn et al., 2023; Du et al., 2023; Madaan et al., 2023) are different. We adhere to the prompts from the source papers to replicate their experiments. So our prompts in these experiments are also different.


**2. Summary of main revisions**

We highlight the main revisions in blue within the paper.

**2.1. Additional experiments**

We conducted additional experiments with GPT-4-Turbo (gpt-4-1106-preview in the table) and Llama-2-70B, employing the settings suggested by the reviewers. The results consistently show that the models’ performance decreases after self-correction. We added the results and full error analysis in Appendix B.2 of our revision.

|                          |                      | # calls | GSM8K | CommonSenseQA |
|--------------------------|----------------------|-------|-------|---------------|
| gpt-4-1106-preview       | Standard Prompting   | 1     | 91.5  | 84.0          |
|                          | Self-Correct (round 1) | 3   | 88.0  | 81.5          |
|                          | Self-Correct (round 2) | 5   | 90.0  | 83.0          |
| Llama-2-70b-chat-hf      | Standard Prompting   | 1     | 62.0  | 64.0          |
|                          | Self-Correct (round 1) | 3   | 43.5  | 37.5          |
|                          | Self-Correct (round 2) | 5   | 36.5  | 36.5          |


**2.2. Writing & diagram**

We adjusted some of our writing and added more explanations in response to the reviews. Additionally, we included diagrams of our analysis on GPT-4 in Figures 1 and 9.

We believe we have addressed each reviewer's concerns and questions in our individual responses. Please let us know if you have any other questions. We look forward to discussing with the reviewers to make the paper even better.

---

### Meta-Review · Area_Chair_smU5 · 2023-12-10

**Metareview:**

This work examines self-correction in LLMs for reasoning without external feedback, and finds LLMs struggle to self-correct or even lead to degraded results. The comments are overall positve where reviewers think self-correction in LLMs is an important topic, and the distinction between intrinsic self-correction and self-correction with external feedback is interesting. Main concerns include: 1. some arguments/claims are not precise or without serious support; 2. the prompts designed for self-correction lack careful consideration (indeed prompts might provide external signals/feedbacks/biases, conflicting with the assumption of "intrinsic" self-correction). AC agrees this study is timely and the problem deserves more systematic and rigorous studies, and thus recommends acceptance as poster.

**Justification For Why Not Higher Score:**

Some arguments and claims could be more precise and backed up with more serious analysis.

**Justification For Why Not Lower Score:**

The problem of self-correction in LLMs is important, and the study in this work is timely and expected to intrigue more systematic and rigorous studies.

---

### Decision · Program_Chairs · 2024-01-16

Accept (poster)